# Are We Moving the Needle for Patients with *TP53*-Mutated Acute Myeloid Leukemia?

**DOI:** 10.3390/cancers14102434

**Published:** 2022-05-14

**Authors:** Rory M. Shallis, Jan P. Bewersdorf, Maximilian F. Stahl, Stephanie Halene, Amer M. Zeidan

**Affiliations:** 1Section of Hematology, Department of Internal Medicine, Yale University School of Medicine and Yale Cancer Center, New Haven, CT 06520, USA; rory.shallis@yale.edu (R.M.S.); stephanie.halene@yale.edu (S.H.); 2Division of Hematologic Malignancies, Department of Medicine, Memorial Sloan Kettering Cancer Center, New York, NY 10065, USA; bewersdj@mskcc.org; 3Department of Medical Oncology, Dana-Farber Cancer Institute, Boston, MA 02215, USA; maximilian_stahl@dfci.harvard.edu

**Keywords:** acute myeloid leukemia, AML, leukemia, p53, *TP53*

## Abstract

**Simple Summary:**

*TP53*-mutated acute myeloid leukemia (AML) represents one of the most informative examples of adverse risk AML. As the currently available therapies have not translated to meaningful advances in the survival of these patients, a clinical trial should be the recommendation for all newly diagnosed patients. CD47/SIRPα axis and TIM-3 inhibition appear to be some of the more promising strategies, but other agents with novel mechanisms of action are in development. We review the pathobiology of *TP53*-mutated AML, the possible heterogeneity among patients with this disease and how some of the novel and emerging therapies may fit into the treatment landscape in the hopefully not-so-distant future.

**Abstract:**

The currently available therapeutic options for patients with *TP53*-mutated acute myeloid leukemia (AML) are insufficient, as they translate to a median overall of only 6–9 months, and less than 10% of patients undergoing the most aggressive treatments, such as intensive induction therapy and allogeneic hematopoietic stem cell transplantation, will be cured. The lack of clear differences in outcomes with different treatments precludes the designation of a standard of care. Recently, there has been growing attention on this critical area of need by way of better understanding the biology of *TP53* alterations and the disparities in outcomes among patients in this molecular subgroup, reflected in the development and testing of agents with novel mechanisms of action. Promising preclinical and efficacy data exist for therapies that are directed at the p53 protein rendered dysfunctional via mutation or that inhibit the CD47/SIRPα axis or other immune checkpoints such as TIM-3. In this review, we discuss recently attractive and emerging therapeutic agents, their preclinical rationale and the available clinical data as a monotherapy or in combination with the currently accepted backbones in frontline and relapsed/refractory settings for patients with *TP53*-mutated AML.

## 1. Introduction

*TP53* is a critical tumor suppressor gene located on chromosome 17p13.1 that encodes the p53 protein, which, in response to cellular stress, including deoxyribonucleic acid (DNA) damage, increases in level and ultimately induces the transcription of the genes responsible for DNA damage repair and cell cycle arrest/apoptosis, among others [1]. As a result, deficiency in the functional p53 protein predicted by mutations in or deletions of this “guardian of the genome” allows cells that would otherwise be destined for programmed cell death (apoptosis) to escape it and foster progression of the malignant disease.

Among patients with a new diagnosis of acute myeloid leukemia (AML), at least 10% will have disease-harboring mutations in *TP53* (*TP53*m-AML) but up to 30% in certain subpopulations such as those with secondary AML, therapy-related AML (t-AML) or acute erythroid leukemia (and nearly all patients with the pure erythroid leukemia subtype) [2,3,4,5,6]. Stemming from the grave deficits in cellular regulation imparted by mutations in *TP53*, *TP53*m-AML is associated with a limited response to traditional AML-directed therapy and poor overall survival (OS). With the currently available therapies, the median OS of a patient with newly diagnosed *TP53*m-AML is approximately 6–9 months, and only approximately 10% of patients will be alive three years after allogeneic hematopoietic stem cell transplantation (alloHCT) [7,8,9,10,11]. As such, the presence of *TP53* mutations has been included in the adverse risk category in the 2017 European LeukemiaNet recommendations [12]. The poor outcomes observed with this subgroup of AML has prompted the development and study of novel agents and combinations to address this critical area of need.

## 2. Mechanisms and the Landscape of *TP53* Alteration 

Alterations of TP53 may occur via several mechanisms, including *TP53* mutations and chromosomal aberrations imparting aberrant protein function and loss of TP53, respectively. All classes of *TP53* mutations have been reported in patients with *TP53*m-AML, but nearly all occur in the DNA-binding domain (encoded by exons 5–8), and the vast majority will be missense mutations, which occur in 80–90% of cases [13,14,15]. Approximately 5–10%, 2–5%, 2–5% and 1–2% of mutations are found to be splice site, frameshift, nonsense and indel variants, respectively [13,14,15]. *TP53* mutations in AML classically involve arginine residues and occur at hot spots (codon positions 175, 220, 245, 248 and 273), specifically R175H, Y220C, R248Q and R273C. P72R mutations outside of the DNA-binding domain are also recurringly identified [2,14,15]. These lesions induce conformational changes in the TP53 protein or induce degradation of the DNA-binding domain that mostly result in a dominant-negative effect, in which the remaining wild-type allele is impaired by the product of the mutated allele, allowing for a selection advantage of the affected clones exposed to cellular stress [16]. Although mutations in *TP53* are largely loss-of-function variants, some predict a partially functional protein [17,18], and others, such as those involving R282, are gain-of-function variants [19]. 

*TP53*m-AML is more likely than the *TP53* wild-type AML to harbor complex karyotype (>3 chromosomal abnormalities), which is namely detected in up to 90% of cases of t-AML [20,21,22]. An increased rate of monosomy 17/abnormal 17p, monosomy 7 and monosomy 5, each found in about 70% of cases, is observed [21,23]. However, for unclear reasons, the rate of classical AML driver mutations (found in approximately 30% of *TP53* wild-type cases) is low, with only 2–7% as cases of *TP53*m-AML-harboring mutations in *NPM1* or *FLT3* [10,21,23,24,25]. 

The loss of band 17p13.1 on which *TP53* is located, either by del(17p) or monosomy 17, leads to an allelic and functional loss of the *TP53* allele. Indeed, AML with del(17p)/monosomy 17 is associated with a median OS similar to AML harboring a *TP53* mutation, and these two lesions should be considered the same for the purposes of risk assessment [26]. In addition, the TP53 protein can be rendered dysfunctional via the overexpression of its chief negative regular, murine double minute 2 (MDM2) [27,28].

The clinical impact of the *TP53* alteration in AML/MDS depends on whether the allelic disruption is monoallelic or biallelic, which determines the amount of functional TP53 protein present. Elegant analyses of patients with *TP53*-mutated MDS demonstrated that approximately 40% of the population harbors disease with a copy-neutral loss of heterozygosity, which, based on the predicted absence of the functional TP53 protein, was significantly associated with inferior survival; conversely, patients with a monoallelic loss of *TP53* fared similar to patients with *TP53* wild-type disease [29]. However, less stringent data support this effect on survival among patients with *TP53*m-AML, which is affected by biallelic TP53 loss in 55–75% of cases [21,29,30,31]. Lastly, the inability to firmly establish copy number alterations or assess the loss of heterozygosity with the current “standard” techniques leaves the prediction of *TP53* biallelic loss to surrogates such as the detection of dual *TP53* mutations, concurrent chromosome 17/17p abnormality or high mutant VAF (i.e., >50%), which have limitations when applied to *TP53*m-AML. 

## 3. Current and Insufficient Standards-of-Care

Intensive, multi-cytotoxic agent regimens, typically comprised of cytarabine and an anthracycline, have been the mainstay of AML-directed therapy for decades. Unplanned subgroup analyses and post hoc evaluations have demonstrated that the rate of complete remission (CR)/CR with incomplete count recovery (CRi) for patients with *TP53*m-AML after cytarabine–anthracycline therapy, either classical “7 + 3” or the liposomal formulation CPX-351, is 20–40% [8,21,23,32,33]; this is in contrast to the 80% rate of CR that is observed for patients with the *TP53* wild-type disease (Table 1) [21,23,24,32]. 

In addition to the inherently poor rates of remission, there is a clear disconnect between the rates of remission and long-term survival for patients with *TP53*m-AML, as an improvement in the former does not consistently translate into an improvement in the latter. However, remission is the first hurdle that must be jumped to ultimately improve long-term outcomes. Definitive consolidation for patients in remission represents the second hurdle. AlloHCT as a consolidative strategy for patients with *TP53*m-AML has been debated, given that only 5–10% of patients will benefit from long-term post-alloHCT OS [13,23,37,44]; however, multivariable analyses have found that alloHCT in the first remission still appears to impart benefits [8,24]. The enthusiasm for proceeding with alloHCT during the first remission likely depends on several factors, including the depth of response to therapy, as estimated by the achievement of molecular remission/mutational clearance.

No intensive therapy is clearly superior to the other in getting patients with *TP53*m-AML into first remission. A randomized, multicenter, phase 3 trial found that CPX-351 demonstrated the superior rates of CR, followed by alloHCT, as well as better median event-free survival (EFS) (2.53 vs. 1.31 months; *p* = 0.021) and median OS (9.6 vs. 5.9 months, *p* = 0.005) when compared with “7 + 3” in older patients with newly diagnosed AML with myelodysplasia-related changes and t-AML [34]. However, post hoc analyses of this trial found that any superiority of CPX-351 appears to be abrogated when specifically evaluating patients with *TP53*m-AML with CR/CRi rates of about 30–40% [9,33]. These figures have been affirmed in multivariable analyses of other retrospective and real-world studies of CPX-351 for *TP53*m-AML [35,46].

The unsatisfactory remission rates observed after intensive therapy must also be viewed in the context of the toxicities with which it is associated. Although no prospective trials restricted to intensively treated patients with *TP53*m-AML are available, unselected patients treated with intensive therapy accept a 10–20% risk of treatment-related mortality by 60 days [34,47]. Advanced age and frailty also increase this risk, with up to 30% and 60% of patients aged >60 years and an Eastern Cooperative Oncology Group (ECOG) performance status >3, respectively, experiencing early mortality [48,49,50,51,52,53,54]. Age is an imperfect surrogate for intensive therapy appropriateness but is often associated with a decreased end-organ reserve and decreased performance status [49,55,56,57,58,59,60]. It is for these reasons that many older patients will be deemed inappropriate to receive intensive therapy and be treated with less intensive, less toxic, disease-modifying therapy. This risk:benefit calculation is further clarified in the context of the dismal outcomes associated with a diagnosis of *TP53*m-AML, and many providers may wish to offer less-intensive therapy to the patient who may actually be appropriate for intensive therapy. 

Azacitidine and decitabine are the standard hypomethylating agents (HMAs) that have been available since the early 2000s and the standard option for the patients inappropriate for intensive therapy [61]. Randomized trials and large, population-based analyses of unselected patients have found azacitidine and decitabine monotherapy to be equally effective in treating AML [62,63]. No differences are apparent when employing indirect comparisons between studies evaluating patients with *TP53*m-AML who appear to have approximately 30–40% rates of CR/CRi with either [10,38,39,40,41,42]. Attempts have been made to replicate the initial promising data for decitabine on a schedule extended 10 days for *TP53*m-AML [43], but no improvement in the remission rates has been realized when compared with classical 5-day decitabine in unselected patients [38,39,40,41,42]. The only randomized, prospective study comparing the two regimens found that 10-day decitabine induced a numerically higher rate of CR/CRi (47% vs. 29%, *p* = 0.40) and better OS (8.5 vs. 5.5 months, *p* = 0.55), but these were not statistically significant (Table 1) [64].

The introduction of venetoclax-inclusive combinations represents a paradigm shift in the treatment of patients with AML who are not appropriate for intensive therapy. 

The randomized phase 3 VIALE-A trial demonstrated that azacitidine + venetoclax was associated with a superior OS when compared with azacitidine monotherapy in mostly older patients with a median age of 76 years (14.7 vs. 9.6 months, *p* < 0.001) [36]. However, patients with *TP53*-mutated AML treated during the initial noncomparative study of azacitidine/decitabine + venetoclax had a 47% rate of CR/CRi [7]; an unplanned and underpowered post hoc subgroup analysis of VIALE-A showed no clear benefit in the OS when evaluating 52 patients with *TP53*m-AML (hazard ratio (HR) = 0.76, 95% CI: 0.40–1.45) [36]. Subsequent retrospective studies have also shown no clear benefit associated with the addition of venetoclax to HMA monotherapy for patient with *TP53*m-AML [65]. The addition of venetoclax to 10-day decitabine has also been studied in *TP53*m-AML and demonstrated a 57% rate of CR/CRi, but the median OS was only 5 months and indistinguishable from that expected with HMA monotherapy [25]. Despite these analogous outcomes in mostly older patients, improved rates of remission may improve the chance of a patient able to proceed to potentially curative alloHCT.

In sum, the optimal frontline regimen for a patient with newly diagnosed *TP53*m-AML is unclear. Intensive regimens and HMA + venetoclax regimens appear to induce the same rates of remission and long-term outcomes. The largest retrospective study of 174 patients with *TP53*m-AML found no difference in the OS between patients treated with azacitidine + venetoclax and intensive therapies [8]. Another retrospective study of 95 patients with *TP53*m-AML found that CPX-351 induction was associated with a better relapse-free survival (RFS) (HR = 0.37, *p* = 0.04) and OS (OR = 0.41, *p* = 0.003), but CPX-351-treated patients were more likely to proceed to alloHCT, invoking a selection bias and the likelihood that patients destined to do better because of less comorbidity and frailty received CPX-351 [66]. It remains unclear whether intensive therapy is the standard for *TP53*m-AML when compared with less-intensive regimens and, among less-intensive regimens, whether venetoclax provides a benefit. Unfortunately, these patients appear to have a near uniform poor prognosis irrespective of the induction strategy chosen. The persistent knowledge gap and poor outcomes make enrollment in clinical trials the unequivocal recommendation for patients with *TP53*m-AML who have the option. 

## 4. Mutant p53 Protein “Refolding” or “Reactivating” Therapies

Several scientific and clinical advancements have provided confidence that *TP53*m-AML may soon be more appropriately treated (Figure 1). Efforts to identify a small molecule capable of restoring a wild-type conformation of the mutant p53 protein identified “p53 reactivation and induction of massive apoptosis” or PRIMA-1, which was noted to do so. Specifically, the structural analog PRIMA-1^met^ or eprenetapropt (APR-246) has demonstrated impressive clinical results consistent with the first molecule to restore functional p53. Eprenetapropt has demonstrated dose-dependent apoptotic effects in the AML cell lines, as well as primary leukemic cells from AML patients [67]. Leukemic cells from patients with *TP53*m-AML, although more chemoresistant, were found to be equally sensitive to the cytotoxic effect of eprenetapropt, which was also found to upregulate the levels of p53 [67]. 

Transcriptome analyses using homozygous *TP53*-mutated myeloid malignancy cell lines and primary AML/myelodysplastic syndrome (MDS) patient samples have demonstrated that even low doses of eprenetapropt can reactivate the p53 pathway and apoptotic programs when administered in combination with a HMA [68]. Cytotoxicity and apoptosis assays have confirmed that eprenetapropt alone is unable to induce apoptosis and, although HMA monotherapy can induce low levels, a combination HMA + eprenetapropt therapy significantly increases apoptosis; to a lesser extent, eprenetapropt is also able to increase the G0/G1 arrest observed with HMA [68]. A subsequent phase 2 study evaluated eprenetapropt (dosed at the recommended phase 2 dose of 100 mg/kg lean body mass (equivalent to 4500 mg/day fixed dosing) daily for four days each 28-day cycle, as identified in the phase 1b portion of the study) with azacitidine 75 mg/m^2^ on a 7-day schedule in patients with higher-risk MDS or oligoblastic (20–30% bone marrow blasts) AML with mutated *TP53* [15]. In the intention-to-treat analysis of the AML population, the median OS was 10.8 months, with improved outcomes observed in the responding patients [15]. In evaluating the entire study population, the responding patients were noted to have significant reductions in the *TP53* variant allele frequency (VAF), with 38% of all patients experiencing *TP53* molecular remission by next-generation sequencing [15]. However, the results of the primary data cut from the phase 3 trial comparing azacitidine + eprenetapropt vs. azacitidine monotherapy in patients with *TP53*-mutated higher-risk MDS, a biologically similar disease, demonstrated no difference in CR, the primary endpoint, between groups (33.3% vs. 22.2%, respectively; *p* = 0.13). The OS was also similar between the arms [69]. An evaluation of the secondary endpoints with maturing data is ongoing. A recent analysis of the phase 2 trial of azacitidine + eprenetapropt with more than two years of median follow-up demonstrated that patients achieving *TP53* mutation clearance prior to alloHCT had the most favorable survival, with a median OS not yet reached [70]. An ongoing phase 2 trial of up to 12 cycles of azacitidine + eprenetapropt post-alloHCT maintenance for patients with *TP53*-mutated AML/MDS demonstrated a 1-year RFS and OS of 58% and 79%, respectively, in the interim analysis of 33 patients (only 12% of whom had pre-alloHCT mutational clearance) [71]. The triplet combination of eprenetapropt with azacitidine + venetoclax is currently being studied in a single-arm trial (NCT04214860). Interim data for this trial demonstrated that, among 30 evaluable patients, the rate of CR and CR/CRi was 37% and 53%, respectively, with a 4-month median duration of response [72]. These data do not appear very different than that associated with azacitidine + venetoclax doublets [7,25,65] and are derived from a single-arm study with a limited number of patients and follow-up. A randomized study is needed before any strong conclusions can be drawn. 

PC14586 is another oral small molecule “reactivator” of the mutant p53 protein specifically characterized by a Y220C mutation, which constitutes <5% of all mutations in *TP53*m-AML [2]. This agent is currently being studied in a phase 1/2 trial of patients with advanced solid tumors with *TP53* Y220C (NCT04585750) but may soon extend to *TP53*m-AML. Despite these efforts, more recent data has suggested that mutant *TP53* may not be a biomarker of the response to these types of agents. Preclinical data demonstrated that the mutated p53 protein binds to NRF2, an antioxidant transcription factor that ultimately decreases the expression of the cysteine/glutamate antiporter SLC7A11 [73]. As an important negative regulator of ferroptosis, a nonapoptotic cell death mechanism, decreased SLC7A11 may be an appropriate biomarker of the response, which is also associated with a decrease in glutathione and, thus, an increase in reactive oxygen species (ROS) with resultant cellular stress and death [73,74]. Indeed, preclinical experiments on the AML cell lines have shown that eprenetapropt leads to glutathione depletion and the induction of ferroptosis irrespective of the *TP53* status [75]. In sum, these agents may not be mutant p53 “reactivators” or “refolders”, and more study is needed to determine more accurate predictive biomarkers.

## 5. Leveraging the Immune System

The use of alloHCT, specifically the hypothesized “graft versus leukemia effect” exerted by adoptively transferred donor T cells, to consolidate AML in remission is classically regarded as the analog to immunotherapy for solid tumors [76]. The heterogeneous nature of AML and lack of highly specific targets as a whole have made clear success hard to come by with regards to the various forms of immunotherapy. However, recent endeavors have supported optimism when evaluating patients with *TP53*m-AML, specifically.

### 5.1. CD47/SIRPα Axis Inhibition

The transmembrane protein CD47 (“’don’t eat me’ signal”) interacts with macrophage-expressed signal-regulatory protein alpha (SIRPα) to dampen macrophage-mediated phagocytosis and cellular destruction [77,78]. CD47 is commonly expressed on normal human cells but is found to be overexpressed on the surfaces of malignant cells from a number of tumor types, including hematologic malignancies, as well as aged erythrocytes [79,80,81,82]. Consequently, the tumor cell overexpression of CD47 prompts its evasion from innate immunosurveillance. Leukemic HSCs isolated from AML patient marrow and peripheral blood samples have been found to uniformly be enriched for CD47 expression when compared with normal controls; furthermore, increased AML HSC CD47 expression is independently correlated with inferior survival [82,83,84]. Murine studies have demonstrated that treatment with monoclonal antibodies directed against CD47 inhibit the in vivo engraftment of AML HSCs and also restore the physiologic level of AML HSC phagocytosis by macrophages [82,83]. In vitro studies have also shown that treatment with HMA or venetoclax increases the cell surface expression of CD47 but, also, the prophagocytic marker calreticulin (“‘eat me’ signal”), supporting synergy with anti-CD47 agents [85]. Many agents are being developed that consist of the CD47-binding domain of human SIRPα and the Fc region of human IgG1 and/or IgG4, thus blocking the CD47 interaction with macrophage SIRPα and overriding the inhibition of phagocytosis. The exact mechanism by which *TP53*m-AML appears to be more susceptible to CD47/SIRPα axis inhibition is unclear but may be related to the differential expression of the aforementioned prophagocytic in this particular disease subgroup [86].

Magrolimab (Hu5F9-G4), a humanized anti-CD47-IgG4 monoclonal antibody, has demonstrated an early, promising efficacy when administered as a priming/intra-patient dose escalation regimen (1–30 mg/kg weekly) in combination with a standard dose azacitidine for patients with AML [87]. Interim analyses from the phase 1b trial of this combination in 52 patients demonstrated a favorable safety profile and, ultimately, a 56% rate of CR/CRi; when specifically evaluating *TP53*m-AML, for which the study population was enriched, 48% of patients achieved CR and 19% achieved CRi for a composite CR/CRi rate of 67% [87]. The median duration of the response was 9.9 months and median OS was 12.9 months [87]. It should be emphasized that these data stem from a single-arm study with a relatively small number of patients and limited follow-up. A phase 1/2 trial of frontline triplet therapy with magrolimab + azacitidine + venetoclax for unselected patients with AML is underway (NCT04435691). The interim data from this trial demonstrated a CR/CRi rate of 100% in seven evaluable patients with *TP53*m-AML and a 57% rate of measurable residual disease (MRD) negativity by a multiparameter flow cytometric analysis (MFC); five patients (71%) successfully proceeded to alloHCT, and all patients were alive at six months after a median 3.9 months follow-up (Table 2) [88].

Other agents, differing in their Fc isotype and/or their molecular weight, have shown a promise of efficacy in *TP53*m-AML. SRF231, a fully humanized anti-CD47 antibody, has demonstrated an ability to increase phagocytosis in AML cell lines, including the p53-null HL60 cell line, as well as in primary bone marrow samples from patients with AML [96]. SRF231 appears to have no effect on hemagglutination or erythrocyte phagocytosis and is currently being studied in a phase 1 basket trial, including patients with hematologic malignancies (NCT03512340) [96]. Evorpacept (ALX148) has demonstrated its ability to increase tumor cell phagocytosis in *TP53*m-AML cell lines and in several mouse xenograft models, which also demonstrated better survival with combination treatments, namely evorpacept + HMA or evorpacept + HMA + venetoclax [85]. For these reasons, evorpacept is currently being studied in a trial in combination with azacitidine + venetoclax for patients with newly diagnosed AML (NCT04755244) (Table 3). Data from the phase 1 portion of the combination of evorpacept + azacitidine for patients with higher-risk MDS demonstrated marrow CR in one out of five patients with *TP53*-mutated disease; the phase 2 portion is ongoing [97].

Anti-CD47 agents with emerging data might soon apply to patients with *TP53*m-AML. TTI-621 and TTI-622 are human recombinant soluble fusion proteins that only minimally bind to normal erythrocytes with reduced transient hemolysis, which is observed with other anti-CD47 agents and has less interference with crossmatching [98,99,100]. The TTI-621 treatment of primary samples from patients with AML amongst patients with MDS and other lymphohematopoietic malignancies led to an increase in macrophage-related phagocytosis in 97% of samples. Murine AML xenograft models have demonstrated similar antitumor activity [98]. The interim data from the first-in-human phase 1 study (NCT02663518) of TTI-621 administered IV weekly in a basket hematologic malignancy trial that included 20 patients with AML determined the maximum tolerated dose (MTD) to be 0.2 mg/kg after an episode of transient grade 4 thrombocytopenia, with 20% of patients experiencing grade >3 thrombocytopenia [101]. None of the 20 patients with AML achieved remission, although one patient in CRi with MRD positivity by next-generation sequencing at the time of enrollment achieved MRD negativity [101]. The absence of a strong clinical efficacy demonstrated in this trial must be tempered by the possibility that the established MTD was inaccurate based on transient thrombocytopenia, as well as the growing knowledge regarding the synergy with combination therapy, such as that with HMA, venetoclax and/or other leukemia-directed therapy, which also provide the mandatory “eat me” signals for efficacy. Notably, TTI-622 has been well-tolerated with dosing up to 18 mg/kg, with both improved pharmacokinetics and objective responses in the ongoing trial with lymphoma [102]. Given that combination therapy is required to optimize the efficacy, TTI-622 is the preferred SIRPα fusion protein to proceed in phase 1b/2 testing for patients with AML, including *TP53*m-AML. An ongoing trial will evaluate TTI-622 in combination with azacitidine for patients with *TP53*m-AML and in combination with azacitidine + venetoclax in patients with *TP53* wild-type AML, both starting at a TTI-622 dose of 8 mg/kg (NCT03530683) [102]. Lemzoparlimab is another anti-CD47 agent currently being dose-escalated in an ongoing trial of patients with relapsed/refractory AML and MDS (NCT04202003). As of the last data cutoff, five patients with AML were enrolled at up to 10 mg/kg without a MTD and, specifically, no dose-dependent hematologic toxicities, although one patient developed grade 3 thrombocytopenia [103]. A CD47 receptor occupancy of 85% was achieved at 10 mg/kg, supporting the likelihood of tumor cell phagocytosis and antileukemia activity. Indeed, one patient treated at 1 mg/kg achieved a morphologic leukemia-free state (MLFS) after two cycles of therapy [103]. Lemzoparlimab is also being studied in an ongoing phase 1 study in combination with azacitidine + venetoclax in patients with newly diagnosed AML/MDS (NCT04912063). Other products being studied in AML include AK117 (NCT04980885), DSP107 (NCT04937166), SL-172154 (NCT05275439) and IBI188 (NCT04485052) (Table 3).

### 5.2. Immune Checkpoint Inhibition and Other Immunotherapy

In addition to its previously described functions, *TP53* influences the induction of interferon-α and -β [104,105,106]; other proinflammatory cytokines (e.g., IL-1, IL-6, IL-12 and TNF) production [107,108] and immune checkpoint regulation; tumor cell *TP53*-dependent PD-1 and PD-L1 upregulation are observed as a response to DNA damage and other genotoxic stress [109,110,111]. This immune checkpoint modulation is hypothesized to be the consequence of several *TP53*-regulated microRNAs (miRs), such as miR-145, the miR-17-92 cluster and miR-34 [112,113,114,115]. *TP53* transcriptionally targets miR-34, which has been shown to directly bind the 3′ untranslated region of *CD274* or the gene encoding PD-L1 [115]. Mutations in *TP53* might also prompt genetic instability, which prompts the generation of other neoantigens that influences the immune homeostasis within the tumor microenvironment. Indeed, a recent transcriptomic analysis of 10,817 tumor aliquots of over 30 tumor types from The Cancer Genome Atlas identified mutations in *TP53* as a predictor of an increased tumor-infiltrating leukocyte fraction, particularly interferon-γ dominant subtypes and a Th2 cell bias to the adaptive immune infiltrate [116]. When compared with *TP53* wild-type AML/MDS bone marrow mononuclear cells, those with *T*P53 mutations exhibited increased PD-L1 expression, as well as the downregulation of miR-34 [117]. However, no convincing clinical data exist to suggest that PD-1 or PD-L1 inhibition is effective in patients with *TP53*m-AML.

Recent attention has turned to alternative immune checkpoint molecules. T-cell immunoglobulin and mucin domain-containing 3 (TIM-3) is a T-cell-negative regulator that is expressed on both CD4+ helper T cells and interferon-γ-secreting CD8+ cytotoxic T cells [118,119]. TIM-3 is constitutively expressed on innate immune cells, such as monocytes/macrophages, natural killer cells and dendritic cells; this key negative immune regulator is also enriched on FoxP3+ regulatory T cells [120,121]. Similar to the PD-1/PD-L1 axis, leukemic blasts in AML have been found to overexpress TIM-3, which ultimately inhibits CD8+ T-cell recognition and, thus, the destruction of these malignant cells [118,119,122,123]. The interim data from an ongoing phase 1 trial of the TIM-3 inhibitor sabatolimab (MBG453) in combination with azacitidine demonstrated CR/CRi in 2 out of 5 patients with *TP53*m-AML and an ORR of 71% (10 out of 14) in patients with *TP53*-mutated higher-risk MDS [91]. The median duration of response in the latter population was 21.5 months (95% confidence interval: 6.7—not evaluable) but with a small sample size of 11 patients [91]. These data provide confidence that this combination might be effective for *TP53*m-AML. A single-arm triplet trial of sabatolimab + azacitidine + venetoclax is underway (NCT04150029). Other early phase clinical trials of sabatolimab (MBG453) in combination with HMA therapy and/or BCL-2 inhibitors for patients with AML/MDS are also currently underway (NCT04266301, NCT03946670 and NCT03940352) (Table 2 and Table 3).

Exploratory analyses from a trial of flotetuzumab, an investigational CD123 × CD3 bispecific dual-affinity retargeting antibody therapy, in AML demonstrated that *TP53*-mutated primary bone marrow samples exhibit greater CD8+ T-cell infiltration and inflammatory cytokine levels [92]. *TP53*-altered AML marrow samples are enriched for interferon-γ and IL-17/tumor necrosis factor signaling programs by gene expression profiling [92]. Intermediate-to-high baseline immune infiltration, a higher tumor inflammation signature and higher interferon-γ levels were predictive of the response to flotetuzumab; in contrast, although *PD-L1* and markers of CD8+ T-cell exhaustion and senescence-like *CD244*, *EOMES*, *LAG3* and *PTGER4* were overexpressed in the *TP53*-mutated population, they did not predict the response to flotetuzumab [92]. These data suggest that the T-cell engager-derived inflammatory tumor microenvironment in AML is not dampened by the presence of *TP53* alterations. Additionally, the observation that protracted interferon-γ signaling induces a polygenic program that is associated with the resistance to conventional therapies in solid tumors invokes a similar concern for *TP53*m-AML [124]. Other anti-CD123 therapies are in development, such as APVO463, XmAb14045/vibecotamab and SAR440234, among others, but none are currently any further along than the phase 1 stage of development. 

## 6. MDM2–p53 Interaction Destabilization

As an E3 ligase, MDM2 promotes the ubiquitination and proteosome-mediated degradation of p53, which remains at relatively low levels under normal cellular conditions [27]. Cellular stress such as the generation of ROS or DNA damage leads to the phosphorylation of MDM2 and p53, disrupting their interaction and preventing p53 degradation [1,27]. High levels of MDM2 are observed in approximately 30% of AML, with MDM2 expression found to correlate with wild-type *TP53* [28]. Although mutations in *TP53* are largely loss of function, some will predict functional proteins [17,18]. The study of the oral MDM2 inhibitor idasanutlin was met with discouraging results, including in the few patients included with *TP53*m-AML [93,125], and might limit the further study of this agent in AML. However, MDM2 inhibition also indirectly promotes the degradation of MCL1, the BH3 family antiapoptotic protein that increases in response to and promotes resistance to venetoclax-containing therapy and a target of some currently studied inhibitory agents (e.g., AMG176, AMG397, S64315 and AZD5991) [126]. These data provide the rationale for a MDM2 inhibitor combination with venetoclax. The MDM2 inhibitor milademetan + low-dose cytarabine with or without venetoclax is being studied in patients with relapsed/refractory AML (NCT03634228), and siremadlin (HDM201) will be studied in a two-arm study in combination with azacitidine + ventoclax and allowed to enroll with *TP53*m-AML (NCT05155709). Other MDM2 inhibitors such as RG7112 (RO5045337), APG-115, BI-907828 and CGM097 are in development and may soon have data for their efficacy in *TP53*m-AML.

Similar to disruption of the MDM2-p53 interaction, other attempts to leverage the ubiquitin–proteosome system to modulate p53 protein activity have been explored as options to effectively treat *TP53*m-AML. The ubiquitin-like protein neural cell developmentally downregulated 8 (NEDD8) is critical to the activity of the Cullin-RING E3 ligases to which it binds and NEDDylates, eventually promoting the proteosome-medicated degradation of proteins such as Nrf-2 and p27. The NEDD8 pathway also appears to influence the activity of the p53 protein via the NEDDylation of MDM2. MDM2, along with NUB1, promotes the nuclear exportation of monoubiquitinated p53 and, thus, its inactivation. The NEDD8-activating enzyme processes NEDD8, rendering it able to bind to its target substrates, and it is for this reason that NEDD8-activating enzyme inhibition has been studied for *TP53*m-AML. The first-in-class NEDD8-activating enzyme inhibitor pevonedistat was studied in combination with azacitidine in patients with AML, and among the eight evaluable patients with *TP53*m-AML, six achieved CR/CRi/PR (75%), with the majority remaining on the treatment at 10 cycles [95]. Ex vivo experiments using AML cell lines, primary patient samples and patient-derived xenograft models, demonstrated that the combination of pevonedistat + azacitidine enhanced venetoclax-mediated apoptosis via the neutralization of MCL-1, a known mechanism of resistance to azacitidine + venetoclax therapy [126,127]. However, the interim analysis of an ongoing phase 1/2 study of the triplet combination of pevonedistat + azacitidine + venetoclax (NCT04266795) demonstrated CR/CRi in six out of eight (75%) patients with *TP53*m-AML but only a median OS of 9 months, limiting the enthusiasm for this triplet combination beyond what is expected with HMA + venetoclax doublet therapy (Table 2) [128]. In sum, the initial attempts to exploit the preclinical data in support of targeting the MDM2–p53 interaction or the ubiquitin–proteosome system to modulate p53 protein activity have been discouraging, but subsequent iterations guided by improved biomarkers and agents may still hold promise for *TP53*m-AML.

## 7. Conclusions

The standard of care for the treatment of *TP53*m-AML is unknown, but the current options available to providers charged with the care of patients afflicted with this disease are inadequate. A meaningful effort is being put forth to develop and study agents, either as monotherapies or in combination with the currently available regimens, with novel or optimized mechanisms of action to address this critical gap. Clinical data for agents exploiting MDM2–p53 homeostasis, such as MDM2 and NEDD8-activating enzyme inhibitors, have recently been met with disappointment. However, among the most promising are agents that inhibit the CD47/SIRPα axis or other immune checkpoints, such as TIM-3. Agents hypothesized to “reactivate” mutant and dysfunctional p53 proteins are also in development. A better understanding of the mechanisms of *TP53* alterations and the nuances that influence the amount of functional TP53 protein, such as higher fidelity and universal methods or copy number determination, are required to identify the variance within the molecular subgroup of *TP53*m-AML. The full profiling of *TP53* alterations should occur in large databases and in larger clinical trials to fully understand their differences in behavior and where research efforts can be more efficiently directed. With a more sophisticated understanding of *TP53* alteration biology and its determinants of the patient outcomes, future randomized trials employing novel agents could soon establish a true standard of care for *TP53*m-AML, a disease that, to date, has served as the chief example of poor-risk AML.

## Figures and Tables

**Figure 1 cancers-14-02434-f001:**
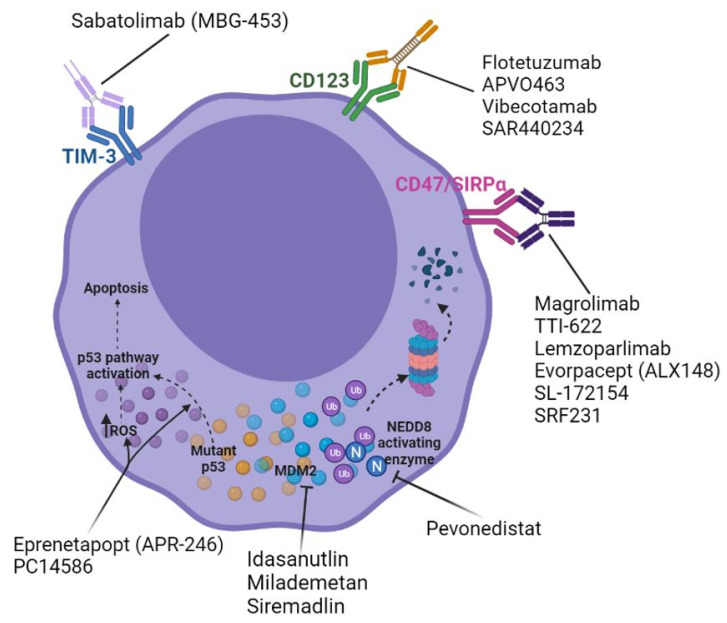
Emerging frontline agents for the treatment of *TP53*-mutated acute myeloid leukemia.

**Table 1 cancers-14-02434-t001:** Summary of the experiences with the currently available frontline therapies for *TP53*-mutated AML.

Regimen	Response Rates	Outcomes	Reference(s)
**Intensive induction therapy**			
Cytarabine + anthracycline (“7 + 3”)	CR: 28–48% CR/CRi: 33–66%	Median EFS: 1.6–5.7 months Median OS: 5.1–6.5 months	[21,23,24,32,33,34]
Liposomal cytarabine + daunorubicin (CPX-351)	CR: 29% CR/CRi: 11–41%	Median EFS: 1.0–8.1 months Median OS: 4.5–8.5 months	[33,34,35]
**Less-intensive induction therapy**			
Azacitidine monotherapy	CR: 40% CR/CRi: 0–40%	Median OS: 7.2 months	[10,36,37]
Decitabine monotherapy(5-day schedule)	CR/CRi: 29%	Median OS: 2.1–5.5 months	[38,39,40,41]
Decitabine monotherapy(10-day schedule)	CR: 31% CR/CRi: 38–47%	Median EFS: 5.7 months Median OS: 4.9–7.3 months	[38,40,42,43]
Azacitidine + venetoclax	CR/CRi: 47–67%	Median EFS: 5.6 months Median OS: 7.2 months	[7,36,44]
Decitabine (5-day schedule) + venetoclax	CR/CRi: 47–50%	Median EFS: 5.6 months Median OS: 7.2 months	[7,44]
Decitabine (10-day schedule) + venetoclax	CR/CRi: 50–69%	Median EFS: 3.4–5.7 months Median OS: 5.2–6.9 months	[25,44,45]

Abbreviations: AML, acute myeloid leukemia; CR, complete remission; CRi, complete remission with incomplete count recovery; EFS, event-free survival; mo, months; OS, overall survival.

**Table 2 cancers-14-02434-t002:** Summary of known clinical data for emerging/novel therapies for *TP53*-mutated AML.

Mechanism of Action	Agent	Interim Clinical Data	References
Mutant p53 “reactivation”	Eprenetapropt (APR-246)	Eprenetapropt + AZA: CR/PR: 36% Median OS: 10.8 months; among patients proceeding to alloHCT with mutational clearance, median OS was not reached Eprenetapropt + AZA + venetoclax: CR: 37%, CR/CRi: 53% Median DoR: 4 months	[15,72,89]
CD47/SIRPα inhibition	Magrolimab (Hu5F9-G4)	Magrolimab + AZA: CR: 48%, CR/CRi: 67% Median DoR: 12.9 months Median OS: 12.9 months Magrolimab + AZA + venetoclax: CR/CRi: 100% (*n* = 7 of 7); 57% MRD-neg Median DoR: 12.9 months	[90,87]
TIM-3 inhibition	Sabatolimab (MBG-453)	Sabatolimab + AZA: CR/CRi: 40% (*n* = 2 of 5)(ORR of 71% in patients with *TP53*-mutated HR-MDS) Median DoR: 6.4 months (21.5 months for patients with HR-MDS)	[91]
CD123 x CD3 bispecific antibody therapy	Flotetuzumab	Flotetuzumab monotherapy (R/R population): CR: 47%, CR/CRi/MLFS/PR: 60% Median OS: 4.5 months (10.3 months among responders)	[92]
MDM2 inhibition	Idasanutlin	Idasanutlin + cytarabine: CR: 4% (*n* = 1 of 25)	[93]
	AMG-232	AMG-232 +/- trametinib: ORR: 0%	[94]
NEDD8 activating enzyme inhibition	Pevonedistat	Pevonedistat + azacitidine: CR/CRi/PR: 75% (*n* = 6 of 8) Most patients remained on treatment by 10 cycles Pevonedistat + azacitidine + venetoclax: CR/CRi: 75% (*n* = 6 of 8) Median OS: 9 months	[11,95]

Abbreviations: CR, complete remission; CRi, complete remission with incomplete count recovery; DoR, duration of response; NEDD8, neural cell developmentally downregulated 8; MDM2, murine double minute 2; MLFS, morphologic leukemia-free state; MRD-neg, measurable residual disease-negative by multiparameter flow cytometric analysis; ORR, overall response rate; PR, partial response; SIRPα, signal-regulatory protein alpha; TIM-3, T-cell immunoglobulin and mucin domain-containing 3.

**Table 3 cancers-14-02434-t003:** Ongoing frontline clinical trials including *TP53*-mutated AML patients without clinical data.

Mechanism of Action	Agent	Regimen	ClinicalTrials.gov Identifier
Mutant p53 “reactivation”	Eprenetapropt (APR-246)	Eprenetapropt + azacitidine + venetoclax	NCT04214860
CD47/SIRPα inhibition	SRF213	Monotherapy	NCT03512340
	Evorpacept (ALX148)	Evorpacept + azacitidine + venetoclax	NCT04755244
	TTI-622	TTI-622 + azacitidine + venetoclax	NCT03530683
	Lemzoparlimab	Lemzoparlimab + azacitidine + venetoclax	NCT04912063
	AK117	AK117 + azacitidine	NCT04980885
	DSP107	DSP107 + azacitidine + venetoclax	NCT04937166
	SL-172154	SL-172154 + azacitidine + venetoclax	NCT05275439
	IBI188	IBI188 + azacitidine	NCT04485052
TIM-3 inhibition	Sabatolimab	Sabatolimab + azacitidine + venetoclax	NCT04150029
MDM2 inhibition	Idasanutlin	Idasanutlin + “7 + 3”	NCT03850535
	Idasanutlin	Idasanutlin + venetoclax (R/R)	NCT02670044
	Milademetan	Milademetan + low-dose cytarabine +/− venetoclax (R/R)	NCT03634228
	Siremadlin (HDM201)	Siremadlin + azacitidine + venetoclax	NCT05155709

Abbreviations: MDM2, murine double minute 2; R/R, relapsed/refractory; SIRPα, signal-regulatory protein alpha; TIM-3, T-cell immunoglobulin and mucin domain-containing 3.

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
