# Peer review of "Are We Moving the Needle for Patients with TP53-Mutated Acute Myeloid Leukemia?"

_cancers, 2022, doi:10.3390/cancers14102434_

Round 1
Reviewer 1 Report
In this review manuscript entitled “Are we moving the needle for patients with TP53-mutated acute myeloid leukemia?”, Shallis et al., reviewed most recent advances in development of novel therapies for TP53 mutant MDS and AML. The manuscript was well-written and almost all significant research papers were included and discussed. The first 3 sections provided strong background information to explain the clinical problems in treatment of TP53 mutant MDS and AML. The remaining sections summarized the observations of the major clinical trials of novel therapies and discussed the issues and potential mechanisms. The major information was very well -organized in tables/figure and discussed in text. Both basic researchers and clinicians will be potential audiences for this review. I only have a few minor concerns.
- Legend is required for the figure.
- Page2 lines71-73: approximately 40% of the population harbors disease with copy-neutral loss of heterozygosity, which both result in absence of functional TP53 protein and predicted inferior survival. This is not clear, because only one situation was described, I don’t understand why both result in…?
- Page 6 line 219 and 220, the redundant title for table 2 need to be deleted.
- Page 8 lines 259-260: …. restore the physiologic level of AML HSC macrophage-mediated phagocytosis. This is confusion, it is difficult to determine which cells mediate the phagocytosis and which cells are engulfed.
Author Response
1. Legend is required for the figure.
RESPONSE: We thank the Reviewer for pointing out this omissions and have now added a figure legend.
2. Page 2 lines71-73: approximately 40% of the population harbors disease with copy-neutral loss of heterozygosity, which both result in absence of functional TP53 protein and predicted inferior survival. This is not clear, because only one situation was described, I don’t understand why both result in…?
RESPONSE: We have clarified the statement to read: “…, which based on the predicted absence of functional TP53 protein was significantly associated with inferior survival.”
3. Page 6 line 219 and 220, the redundant title for table 2 need to be deleted.
RESPONSE: This has been deleted.
4. Page 8 lines 259-260: …. restore the physiologic level of AML HSC macrophage-mediated phagocytosis. This is confusion, it is difficult to determine which cells mediate the phagocytosis and which cells are engulfed
RESPONSE: This sentence has been reworded to state: “…physiologic level of AML HSC phagocytosis by macrophages.”
Reviewer 2 Report
In this review the authors have described in detail about the role of TP53 mutations in acute myeloid leukemia biology, prognosis and therapy. Review is well writing, very informative and covers most of the recent information’s. I have few suggestions
Suggestions: Authors should briefly comment
- on the role of TP53 mutations in therapy-related AML
- on TP53 co-occurring mutations in AML
- on the role of TP53 mutations in AML leukemic stem cells
Author Response
1. Suggestions: Authors should briefly comment on the role of TP53mutations in therapy-related AML, on TP53 co-occurring mutations in AML, on the role of TP53 mutations in AML leukemic stem cells
RESPONSE: We thank the Reviewer for this suggestion. We have cited the seminal Wong et al. Nature 2015 paper illustrating pre-existing TP53-mutated HSC clonal expansion in response to chemotherapy (as well as the paper by Yan et al. Leukemia. 2020). The revised manuscript now also includes an additional paragraph detailing the cytogenetic and molecular landscape of TP53-mutated AML, namely the low rate of co-mutation.
Reviewer 3 Report
Shallis and colleagues submit a well-written summary of current clinical experience with TP53 mutant AML. They briefly describe the biology of TP53 mutant AML and spend the bulk of the manuscript describing relevant clinical outcomes and attempts to improve them. Overall, the review is easy to read, thoughtfully developed, and provides a fairly comprehensive overview of current results. It should be a welcome summary for clinicians and scientists.
Minor suggestions.
Several important clinical questions are raised. The question of whether TP53m patients should receive a transplant is important. As the authors note, these patients as a group do not do well, but in the limited data we have, it appears that TP53m patients who are transplanted fare better than those who do not. The authors may want to expand this discussion a little more given the clinical relevance. Any thoughts or insights would be helpful. Should we focus transplants on patients with molecular remissions?
The question of whether cytarabine-based induction vs HMA induction is associated with better survival is being directly addressed in the InDACtion trial in Europe (NCT02172872). That study might be worth a mention. Results should read out in the near future.
The discussion of HMA/venetoclax is well done and provides the reader with a good sense of the limitations of the data, as well as the apparent limitations of adding venetoclax to HMA in TP53m cases.
TP53 mutations are enriched in erythroleukemia. This might be worth noting, as this could be relevant to the ferroptosis effects of some therapies, like eprenetapropt and the anemia complications of magrolimab.
The observation that MDM2 inhibition could degrade MCL1 is a helpful highlight. This provides an additional option to MCL1 inhibition.
Author Response
1. Several important clinical questions are raised. The question of whether TP53m patients should receive a transplant is important. As the authors note, these patients as a group do not do well, but in the limited data we have, it appears that TP53m patients who are transplanted fare better than those who do not. The authors may want to expand this discussion a little more given the clinical relevance. Any thoughts or insights would be helpful. Should we focus transplants on patients with molecular remissions?
RESPONSE: We are sensitive to the Reviewer’s comment that is of critical importance, but wish to stress that this submission is intended to be part of a special addition relating to novel therapies, which we hope to retain as our focus. In an effort to address the issue raised by the Reviewer, the second paragraph of section 3, including the statement that “…multivariable analyses have found that alloHCT in first remission still appears to impart benefit” and cite Prochazka et al. Haematologica 2019 and the recent and largest report on this population to date from Badar et al. Am J Hematol. 2022. In section 4 we subsequently cite the follow-up data from the phase 2 trial of azacitidine + eprenetapropt presented at this past ASH meeting that is the largest and most informative data set available in support of prioritizing patients that achieve molecular remission for alloHCT. We have added a statement regarding this factor to the end of the paragraph relating to alloHCT in Section 3 (“Current and insufficient standards-of-care): “The enthusiasm for proceeding with alloHCT in first remission likely depends on several factors including depth of response to therapy as estimated by the achievement of molecular remission/mutational clearance.”
2. The question of whether cytarabine-based induction vs HMA induction is associated with better survival is being directly addressed in the InDACtion trial in Europe (NCT02172872). That study might be worth a mention. Results should read out in the near future.
RESPONSE: We agree that this could be an informative trial upon read out, but would note that this trial, although randomized, is neither specific to nor stratifying by TP53-mutated disease, the population of interest.
3. The discussion of HMA/venetoclax is well done and provides the reader with a good sense of the limitations of the data, as well as the apparent limitations of adding venetoclax to HMA in TP53m cases.
RESPONSE: We thank the Reviewer for this praise.
4. TP53 mutations are enriched in erythroleukemia. This might be worth noting, as this could be relevant to the ferroptosis effects of some therapies, like eprenetapropt and the anemia complications of magrolimab.
RESPONSE: We have noted this in the revised manuscript and supported this addition with two references from Rose et al. Leukemia 2017 and Montalban-Bravo et al. Blood 2017.
5. The observation that MDM2 inhibition could degrade MCL1 is a helpful highlight. This provides an additional option to MCL1 inhibition.
RESPONSE: We agree with the Reviewer and have slightly added to the section to illustrate this, also specifically naming some MCL1 inhibitors in development.